# Friend or Foe: How Do Consumers and Producers Affect the ESG Rating Index? Evidence from China's Market of Organic Milk

**Jiangyuan Hou** [1], **Yanping Wang** [2] **and Mingyue Du** [3,*]

1   School of International Trade and Economics, University of International Business and Economics, No. 10 Huixin East Road, Chaoyang District, Beijing 100029, China
2   School of Economics, Henan University of Economics and Law, No. 108 Jinshui East Road, Zhengzhou 450011, China
3   School of Economics, Beijing Technology and Business University, No. 11 Fucheng Road, Haidian District, Beijing 100048, China
*   Correspondence: mydu@st.btbu.edu.cn

**Abstract:** This study attempts to investigate how product attributes and consumer socio-economic status affect organic milk (OM) product ESG performance. There are not many pertinent studies on the interaction between the EGS evaluation system and organic milk products. Thus, this paper develops a targeted hedonic-regression model based on the EPB and ECB theories to investigate the ESG and sub-ESG impact of the above attributes based on the consumption dataset from Kantar Worldpanel. We also introduce the moderating variable of marketing resource intensity (MRI) to check the influencing mechanism of the sub-ESG rating index. This study demonstrates that the intensity of marketing resources has a moderating effect on the ESG performance of organic milk producers. Moreover, we also found evidence that the household's family size and internal product attributes have a more significant impact on the ESG rating index. In addition, compared to social and governance information, the environmental information disclosure index has a more significant influence on consumers' socio-economic status and product attributes compared with the social and governance rating index. This study not only contributes to deepening the understanding of the ESG evaluation system but also provides scientific guidance for the producers' ESG strategy. Therefore, to increase the competitive advantage of OM enterprises, they should endeavor to construct a comprehensive evaluation system of entertainment social governance rating indexes and engage in environmental investment activities.

**Keywords:** ESG; sub-ESG; consumers' socio-economic status; product attributes; China's market of organic milk

## 1. Introduction

It is well known that the objective of businesses is to maximise their profits by producing valued products that adapt to their customers' demands [1]. However, because of an increased awareness of environmental, social, and governance (ESG) issues in the business market, consumers tend to realise the importance of the ESG performance of enterprises. ESG issues, including the three sub-ESG rating indexes of environmental, social, and corporate governance, refer to the assessment of a company's performance and impact in these areas. Analysing the impact of ESG evaluation systems can improve a company's core competencies and identify areas for optimisation, which can ultimately generate long-term value, reduce risk, and have a positive impact on society and the environment. To captivate the hearts of customers and improve their user stickiness, enterprises have eagerly participated in ESG activities. Correspondingly, the ESG performance of enterprises is increasingly essential for enterprises' decisions [2]. It is estimated by Fang et al. [3]

that the number of A-share-listed firms issuing ESG reports has risen from 872 in 2018 to 1130 in 2021. In this context, the issue of how to stimulate more ESG activities has been evaluated by scholars and practitioners [4], which has potentially transformed into the new competitive strategy of enterprises [5,6].

Regarding China's organic milk (OM) market, it has experienced sustained growth since 2005. In 2005, the Certification and Accreditation Administration of the People's Republic of China (CNCA) released a national standard for OIMP called GB19630.1-4-2005 in 2005, which laid the foundation of the market for OM [7]. According to Xu et al. [8], the market share of OM will be expected to reach over 10% in the future market for milk powder in China, a jump from less than 1% in 2007. This is explained by Scarpato et al. [9] as the consumers' demand for sustainable products behind the Corporate Social Responsibility (CSR) motivation for the enterprises, which is closely associated with the ESG performance [2]. In the long term, the ESG activities linked to ESG performance are able to strengthen the word-of-mouth (WOM) of enterprises and brands, generate consumer loyalty [10], and obtain better market performance [2]. Thus, the issue of the CSR impact of the OM product is investigated by scholars from the perspective of consumers' socio-economic status and product attributes.

From the perspective of the former, Youn and Kim [11] and Chen et al. [2] find that the consumers' socio-economic features (e.g., purchasers' gender, their educational level, their income, etc.) are positively associated with the CSR performance of OM products. Carrigan et al. [12] provide evidence that experienced consumers are more concerned about the ethical issue of OM products, which is supported by Perrini et al. [13], further noting that the ethical performance of enterprises could potentially increase consumers' trust in organic foods. Vahdati et al. [14] determine that there is a significant positive relationship between consumers' purchasing attitudes and the ethical consumption of milk products. Regarding the latter, most of the scholars mainly focus on the effect of product attributes (e.g., product taste, product size, product skin, organic label, product origin, etc.) on the CSR performance of milk products [15,16]. In particular, Anselmsson and Johansson [17] find the positive effect of product packaging and the CSR image of OM products, which reveal the quality of branded products.

Previous scholars studied the impact of consumers' socio-economic status and product attributes on CSR performance, but the issue of how relevant variables affect ESG rating indices has rarely been discussed in the economic and business literature. This study aims to fill this gap to some extent, taking organic milk products as an example. Based on the theories of EPB and ECB, this research achieves these targets via the following path. Firstly, this study delves into the mechanisms by which consumers' socio-economic status and product attributes influence the ESG rating index rather than simply analysing its impact on social responsibility performance. Secondly, the moderating variable of marketing resource intensity (MRI) is introduced into our econometric analysis in order to link the marketing tools to the ESG performance of OM producers. Thirdly, the hedonic analysis is conducted for the sub-ESG rating index, and the importance of the environmental disclosure index is argued.

This aim is to investigate the ESG effect of consumers' OPBs on OM. The following issues will be addressed in this paper: (i) how do consumers' socio-economic status and product attributes affect the ESG and sub-ESG performance of OM producers?; (ii) what are the influencing mechanisms of the ESG rating index regarding the above attributes?; and (iii) how do producers carry out the ESG strategy in order to improve their market competitiveness?

Actually, the ESG rating index gives us fresh insights about the measurement of CSR performance, which includes the non-financial movement in the long run [4,5]. The purpose of this study is to examine how consumers' socio-economic characteristics and product attributes affect the ESG performance of OM products and to provide support to companies in constructing a comprehensive system of ESG evaluation. Therefore, our research is novel in various dimensions: On the one hand, we further advance the hedonic model of ESG and

sub-ESG rating indexes by combining the ESG rating index from Bloomberg reports and the Wind Data Service Platform dataset. On the other hand, as proposed by Ullah [18], we introduce the moderating variable of marketing resource intensity (MRI) in order to reveal the influencing mechanism of the ESG impact. In particular, the introduction of the hedonic regression method contributes to the quantitative analysis of the ESG rating index, which can provide scientific guidance for the construction of the ESG evaluation system. The structure of this paper is arranged as follows. Section 2 is the theoretical framework and hypothesis development. The hedonic model is shown in Section 3. Sections 4 and 5 are the descriptions of the data and analyses of the regression results. The discussion, implications, and limitations are drawn out in Section 6.

## 2. Theoretical Framework and Research Hypothesis

For OM products, the following issues need to be addressed in the future: (i) environment: carbon emissions and energy consumption can hurt the environment; organic milk production requires a large amount of water, which is prone to waste and pollution; (ii) society: dairy companies need to ensure product quality and safety, including regulation and control from supply chain management to production processes; and (iii) corporate governance: the disclosure of relevant information by dairy companies, including financial information, corporate governance structures, and transparency of decision-making processes.

Research on CSR offers details on the constructs of ethical product behavior (EPB) and ethical consumption behavior (ECB), corresponding to the demand and supply theory of CSR [19,20]. Kuokkanen and Sun [19] noted that the EPB and ECB potentially increase the sustainability of consumption and production, thereby significantly impacting the CSR performance of consumers and producers. Thus, some scholars define the above conceptions before analysing the ESG impact of socio-economic status and product attributes. Regarding the former, the theory of the ECB is able to describe the moral effect of consumption behavior [21], which Auger et al. [22] call ethical consumerism. ECB is defined by Crane and Matten [23] as "the conscious and deliberate purchasing decision based on moral belief and values", which contributes to a better understanding of consumers' sustainable purchasing behavior of OM products [24,25].

In fact, government agencies can use the ESG evaluation system to ascertain the social performance of the industry and then create a more reasonable and consumer-friendly food production licensing system, thereby accomplishing macro-level market regulation [19]. Corporate managers can develop their upcoming competitive strategies and marketing tools based on the evaluation's results in order to increase their competitiveness and gain long-term advantages [21]. According to the ESG indicators, consumers are capable of choosing goods that match their ethical standards [21].

Based on the theory of the ECB, consumers' purchasing behavior is determined by ethical obligation and CSR [26], which Al-Haddad et al. [27] explained as the moderating effect of consumer participation through social media. Concerning the latter, the theory of EPB links CSR into the enterprises' production behavior, which is defined as "the overall process of ethical product design and production in order to meet the consumers' demand for ethically sustainable products" [19]. The above theories are closely associated with the enterprises' non-financial investment activities. This can be measured by the ESG rating index, which is defined by Clementino and Perkins [28] as "the expert's evaluation of a company based on a comparative assessment of their quality, standard or performance on environmental, social or governance issues". Therefore, the theories of ECB and EPB can be successfully applied to the econometric analysis of ESG performance regarding the OM product.

From the perspective of business management, ESG non-financial investment activities have gradually transformed into their competitive strategy [29], which implies that enterprises are responsible for public interests while seeking economic benefits [30]. Thus, these activities contribute to the establishment of the enterprises' ethical image and the formation of consumer stickiness [2,10]. From the perspective of consumers' socio-economic

status, Carrigan et al. [12] and Tian et al. [31] find that experienced consumers with higher socio-economic features (e.g., larger family size, female consumers with higher income and educational level, etc.) tend to focus on the ethical issue of OM enterprises. Consumers trusting these enterprises have a willingness to pay for the corresponding branded products [32]. This is explained by Ullah [18] as the moderating role of marketing resource intensity (MRI), which is defined as the extent of enterprises' investment in marketing activities including advertising, consumer awareness, and consumer service [33]. Wagner [34] argues that advertising activities form the basis of a long-term cooperative relationship between buyers and sellers. De Boer [35] further contends that the maintenance of consumer awareness and consumer service is able to enhance market communication by sending information about enterprises' sustainable endeavors.

From the perspective of producers, the EPB of producers is closely associated with their ESG performance, which is able to reveal the product quality and affect the consumers' purchasing attitude [25]. Thus, product attributes have attracted increasing attention from consumers [22], which can be explained by the fact that product attributes are closely linked to the corporate association—the reflection of ESG [36]. Through the moderating power of MRI, the positive information of product attributes can be received by ethical consumers, thereby generating differentiated product advantages for OM products and resulting in a higher willingness-to-pay (WTP) for this product [18].

Regarding the sub-ESG performance, Suciu et al. [37] find that consumers' purchasing behavior of OM products has a more significant impact on the environmental friendliness and product safety, which are associated with the Environmental performance of OM products. This is what McEachern and McClean [24] call the environmental influence of buying behavior regarding sustainable-green foods.

From the above literature review, the following hypotheses are proposed by us:

**H1.** The consumers' socio-economic features and product attributes, moderated by MRI, positively affect the ESG rating index;

**H2.** The environmental disclosure index is largely determined by the consumers' socio-economic features and product attributes relative to the Social and Governance disclosure indexes.

Based on EPB and ECB theory, it can be determined that the EGS rating index is a valid response to the performance of organic milk products on environmental, social, and governance issues. During the practical argumentation in the subsequent subsections, the hedonic regression model is first extended to identify the influencing variables of the ESG rating index as the demographic characteristics of consumers and product attributes. These variables are in line with the ECB theory that "consumers consciously choose the products they buy based on their moral beliefs". The further development of an offline segmentation model and the decomposition of sub-ESG indices (environmental, social, and governance) are in line with the EPB theory that "CSR and production activities are based on meeting consumer demands". The results of the data analysis will ultimately show whether the assumptions of the theoretical part are true and whether they are supported by the dataset.

## 3. Research Method

This study aims to examine how consumers' socio-economic characteristics and product attributes affect the ESG performance of OM products, while market research based on consumer demographics can further improve the construction of ESG evaluation systems. Only by understanding the influence mechanisms of the relevant variables can we find effective strategies and diversified marketing approaches to improve the competitiveness of companies. We employ the hedonic regression model in this paper. This method suggested by Lancaster [38] has been widely adopted in the analyses of the price premium of food products [39,40], which is explained by Ladd and Zober [41] as the sum of the marginal weights of the characteristics included in the products multiplied by the marginal intrinsic price premium of product attributes. However, little empirical evidence provides insight

into the person-oriented drivers of the ESG rating index from the hedonic model [42]. Thus, we extend the traditional hedonic regression methodology into the analyses of the ESG rating index by combining it with the retail transaction data of OM offered by CTR Market Research Co. Ltd. In this model, based on the thought of Muller [42], the ESG rating index is considered to be a dependent variable and can be given by:

$$ESG\_B = f(X_c, X_p) = f(X) + \varepsilon \tag{1}$$

where *ESG_B* and $\varepsilon$ denote the ESG rating index offered by Bloomberg reports and error terms, respectively. $X_c$ and $X_p$ are the influence factors of consumers' socio-demographic characteristics and product attributes, respectively, which are listed in Table 1. In detail, based on the thought of Ankamah-Yeboah et al. [43], we adopt the log-linear model specification to measure the ESG effect of the above-mentioned influence attributes:

$$\ln(ESG\_B) = \beta_0 + \sum_i \gamma_i X_{ci} + \sum_j \alpha_j X_{pj} + \theta_{place} + \theta_{year} + \varepsilon \tag{2}$$

where $\ln(ESG\_B)$ is the logarithmic form of the ESG rating index from 2005 to 2020, and $\gamma_i$ and $\alpha_j$ represent the estimated coefficients of the consumers' socio-economic features variables ($X_{ci}$) and product attributes ($X_{pj}$), which are able to estimate the implicit ESG effects. The subscripts i and j are the maximal numbers of the above variables. Note that, to decrease the possible regression biases caused by the heterogeneity of purchase regions and years, we control for the effect of the corresponding variables.

**Table 1.** The definitions of regression variables.

| Variables | VT | Definitions |
|---|---|---|
| ESG_W | Cont | Wind-ESG rating index. |
| ESG_B | Cont | Bloomberg-ESG rating index. |
| Fs | Cont | The family size of households. |
| w | Cont | The consumers' wage. |
| Gender | Dum | Female = 1; otherwise = 0. |
| Age | Cont | The consumers' age. |
| Edu | Dum | Bachelor degree = 1; otherwise = 0. |
| Pack | Dum | Canned package = 1; otherwise = 0. |
| Vers | Dum | General version = 1; otherwise = 0. |
| Fla | Dum | Plain flavored = 1; otherwise = 0. |
| Func | Dum | Ordinary = 1; otherwise = 0. |
| Fat | Dum | Including fat = 1; otherwise = 0. |
| Sugar | Dum | Including sugar = 1; otherwise = 0. |
| Probio | Dum | Including probiotics = 1; otherwise = 0. |
| Kids | Dum | Designed for kids. |
| MRI | Cont | Marketing resource intensity. |

Note: S.D., VT, Cont, and Dum are abbreviations for standard derivation, variable type, continuous variable, and dummy variable, respectively. In particular, the unit of income is CNY 1000.

In addition to the model of the ESG rating index, we also estimate a split model of the online channel and the traditional offline channel to compare the ESG effects in the multi-channel environment. Thus, the regression model of the ESG rating index concerning the above explanatory variables is given by:

$$\ln(ESG\_B_m) = \beta_{0m} + \sum_{im} \gamma_{im} X_{cim} + \sum_{jm} \alpha_{jm} X_{pjm} + \theta_{place} + \theta_{year} + \varepsilon_m \; if \; m = k \; (k = 0, 1) \tag{3}$$

where the subscript m denotes the consumer's purchasing channel. Particularly, *k* = 0, 1 corresponds to the traditional offline channel (*k* = 0) and online channel (*k* = 1), respectively.

In order to check the robustness of the baseline regression, we replace the explanatory variable ESG_B with the alternative ESG evaluation index from the Wind Data Service

Platform dataset (*ESG_W*) [44], which is collected from company resources such as CSR reports, annual reports, proprietary Bloomberg surveys, etc. [3].

Moreover, the ESG performance of food products hinges on the development of food quality supervision management [45,46]. To measure the policy effect of the ESG rating index, we view the food production license policy as a natural experiment that can reveal how consumers' socio-economic status and product attributes impact the ESG performance of OM producers in the long term. Note that, in 2013, the National Medical Products Administration (NMPA) issued the "Rules for the Examination for Licensing Criteria for Enterprises Producing Formula Milk Powder for Infants and Young Children (2013 Version)" aimed at promoting the sustainable advancement of infant milk powder, which stands for the construction of the food production licensing (FPI) system (http://epaper.bjnews.com.cn/html/2016-06/14/content_639401.htm?div=-1 (accessed on 1 July 2023)). Based on this standard, the number of infant milk powder (IMP) enterprises decreased from 133 to 51 in China, and the OM products from Ireland, Australia, Denmark, Germany, France, the Netherlands, and New Zealand are selected for the first registration list of overseas IMP producers by NMPA.

In fact, FPI plays an essential role in the formation of the ESG evaluation system. On the one hand, the new policy contributes to advancing the professional standard of the IMP industry, which potentially normalizes the production behavior of IMP firms. On the other hand, the standardisation of the IMP production process can improve the quality of this product. This is positively linked to the buying experience, which helps rebuild market confidence.

Thus, the following regression is adopted to examine the long-term policy effect of the ESG rating index:

$$\ln(ESG\_B) = \mu + \sum_i \lambda_i \cdot X_{ci} + \sum_j \lambda_j \cdot X_{pj} + \theta_{place} + \theta_{year} + \varepsilon \ if \ post = t, \ t \in (0,1) \quad (4)$$

where post equals one if the examined year is more than or equal to 2013, and zero otherwise; $\varepsilon$ is the residual term of the regression model. We further compare the estimated coefficients of the above explanatory variables in a multi-channel context.

Then, to measure the sub-ESG effect of OM, as proposed by Bloomberg, we further divide the ESG rating index (ESG_bloom) into the following components: Environmental Disclosure index (E_index), Social Disclosure index (S_index), and Governance Disclosure index (G_index). In particular, due to the moderating influence of MRI offered by the WorldScope dataset on CSR [18], we introduce the variable of MRI into the hedonic regression, which can be given by:

$$\begin{aligned} \ln(ESG\_B_s) &= \beta_{0s} + \sum_i \gamma_{is} X_{cis} + \sum_j \alpha_{js} X_{pjs} + \sum_i \delta_{is} \cdot MRI \cdot X_{cis} \\ &+ \sum_j \varphi_{js} \cdot MRI \cdot X_{pjs} + \theta_{place} + \theta_{year} + \varepsilon_s, \ s \in \{1,2,3\} \end{aligned} \quad (5)$$

where s ∈ (1,2,3) represents the Environment index (E_index), Social index (S_index), and Governance index (G_index), respectively. In Equation (5), the interaction terms of MRI (from the fourth term to the fifth term) correspond to the coefficients $\delta$ and $\varphi$, which can evaluate the moderating impact of MRI on consumers' social and socio-economic features and product attributes, respectively.

## 4. Data

The primary data source comes from Kantar Worldpanel (KWP), provided by CTR Market Research Co. Ltd. in China. This representative panel data contains organic milk (OM) transactions for over 8000 households from 2005 to 2020 in China. For every OM product purchased by consumers, available information contains the purchasing time and location, the quantity bought and paid, the consumers' socio-demographic characteristics, and product attributes. In addition, to examine the hedonic analysis of the ESG rating

index, we introduce the variables of the ESG rating index from the Wind Data Service Platform and Bloomberg reports, which are specified as ESG_W and ESG_B in this paper. Note that, as one of the first qualified foreign-related survey agencies accredited by the National Bureau of Statistics in China, CTR Market Research Co. Ltd. is committed to combining nearly 30 years of China's market insight experience with Internet Big Data technology to offer comprehensive market analysis. Thus, the reliability and validity of this transaction data are guaranteed. Data are often provided by market research business centers in China but is often commercially confidential, so comprehensive and detailed data is not available. The availability of data can also be limited by geographical area and time of access. The data needs to be matched to the region or market being analysed, and there is a need to be aware of timeliness to understand current competitive trends.

In addition, the change in production in China between 2005 and 2020 was obtained by consulting the China National Database.

As shown in Figure 1, organic milk production is on an upward trend, with a more significant rise during the 2019–2020 COVID-19 pandemic. Due to the long-term home-based office status and the awakening of health and hygiene awareness [40], the organic milk industry has achieved a 14.7% increase in sales in China, accounting for 29% of consumption expenditure at home but only 10% of consumption expenditure away from home.

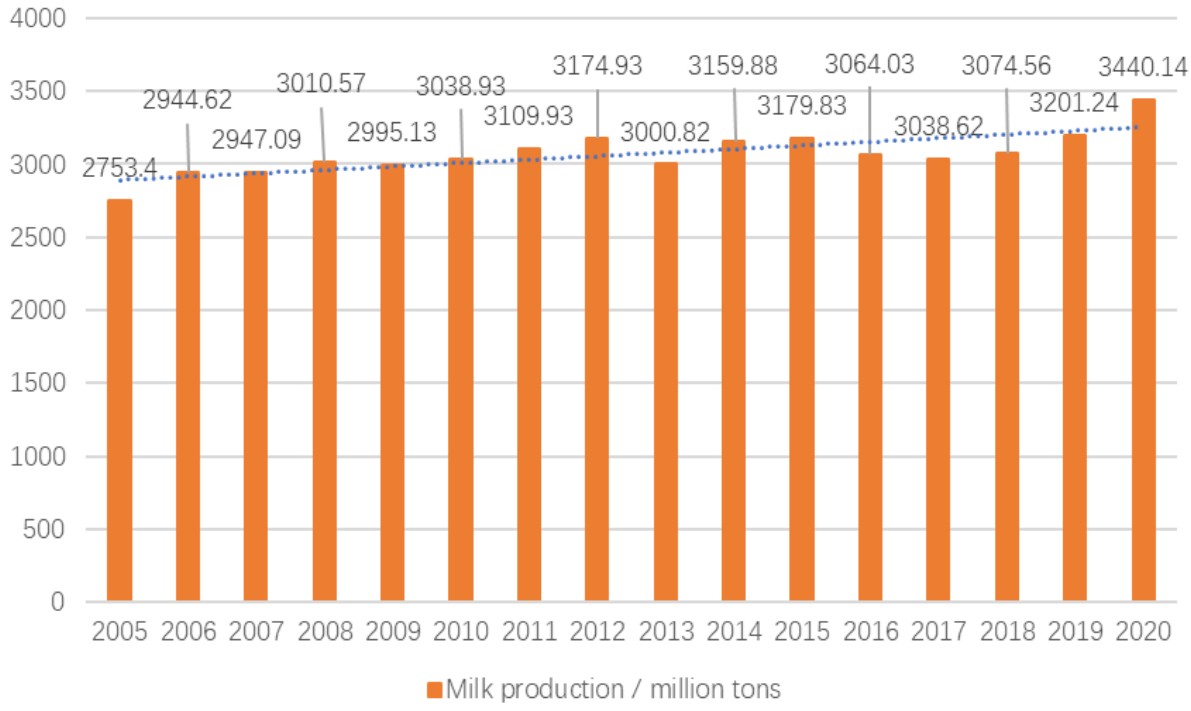

**Figure 1.** China's organic milk production, 2005–2020.

For each OM transaction, the consumers' socio-economic characteristics (CSC) includes family size (*fs*), the consumers' wage (*w*), the purchasers' gender (*gender*), the purchasers' age (*age*), and their educational level (*edu*). Product attributes in our dataset include packaging (*pack*), version (*vers*), flavor (*fla*), function (*func*), fat content (*fat*), sugar content (*sugar*), probiotics (*probio*), and design for kids (*kids*). We define the above product attributes in this section. In this section, we define the product attributes of OM based on the above specifications. The product package is defined as the extrinsic packaging of OM products, including the can, box, barrel, and pop-top can. The product version is specified as whether the OM product is produced in the foreign milk source base and imported in specific regions, which includes the general version (Chinese-foreign joint-brand) and the special version named after the sale's region (e.g., personalised version for China, Germany, the US, Australia, the UK, France, and the Netherlands, etc.). The attribute of flavor contains plain milk, camel milk, goat milk, and other flavours of milk products, which are specified as the

animal sources of OM products. Moreover, the product function can measure whether OM products have been produced by advanced technology (e.g., ordinary OM, immune OM, modest hydrolysis OM, and anti-allergic OM), which is specified as the product's chemical properties. In addition, we define the attributes of fat content, sugar content, and probiotics as having whole fat, sugar, and probiotics, respectively. In particular, the variables of packaging and version describe the external features of the OM product. Correspondingly, the rest of the product attributes mainly measure the internal characteristics of this product.

Summary Statistics. The descriptive statistics for variables used in this hedonic regression analysis are shown in Table 2. From this table, the mean values of the ESG rating index from the Wind Dataset and Bloomberg reports are 4.653 and 4.639, respectively. This shows that the ESG rating index has the most decisive impact on the competitive strategy of enterprises [2]. We can also notice that the mean value of the ESG rating index in the online environment is larger than the corresponding value in the offline channel, which implies that the ESG rating index plays a more significant role in the business performance of OM enterprises. Regarding the consumers' socio-economic status, the mean values of family size, income, age, and gender variables (3.326, 2.678, 25.956, and 0.479) demonstrate that young female consumers with one child and higher income have a strong preference for OM products, which Gao et al. [47] explained as "the potential cross-generation concern about sustainable development". Particularly, the mean value of the *edu* variable is 0.653, showing that consumers with higher educational levels are willing to pay for OM products. This is in line with the findings of Dimitri and Dettmann [48] and Schröck [49]. Note that the mean value of internal product attributes is relatively smaller than that of external attributes, confirming the power of OM product quality for the ESG performance of enterprises [50].

**Table 2.** Summary of statistics for regression variables.

| Variables | Total Sample (N = 64,618) | Online Sample (N = 35,104) | Offline Sample (N = 29,514) |
|---|---|---|---|
| | Mean (S.D.) | Mean (S.D.) | Mean (S.D.) |
| ESG_W | 4.653 (2.127) | 4.630 (2.156) | 4.618 (2.210) |
| ESG_B | 4.639 (2.101) | 4.616 (2.049) | 4.610 (2.130) |
| Fs | 3.326 (0.781) | 3.185 (0.681) | 3.084 (0.826) |
| w | 2.678 (2.011) | 2.594 (1.978) | 2.369 (2.017) |
| Gender | 0.479 (0.501) | 0.519 (0.489) | 0.477 (0.499) |
| Age | 25.956 (9.296) | 23.983 (9.258) | 25.880 (9.317) |
| Edu | 0.653 (2.127) | 0.529 (2.155) | 0.448 (2.211) |
| Pack | 0.243 (0.462) | 0.224 (0.430) | 0.215 (0.476) |
| Vers | 0.178 (0.267) | 0.177 (0.257) | 0.102 (0.268) |
| Fla | 0.950 (0.278) | 0.941 (0.235) | 0.955 (0.246) |
| Func | 0.915 (0.261) | 0.908 (0.289) | 0.921 (0.292) |
| Fat | 0.502 (0.384) | 0.505 (0.401) | 0.530 (0.500) |
| Sugar | 0.499 (0.275) | 0.455 (0.500) | 0.495 (0.599) |
| Probio | 0.863 (0.343) | 0.761 (0.345) | 0.864 (0.342) |
| Kids | 0.214 (0.473) | 0.207 (0.491) | 0.419 (0.593) |
| MRI | 1.651 (2.145) | 1.661 (2.214) | 1.642 (0.482) |

The second and third columns of Table 2 show the mean value of explanatory variables regarding the ESG rating index. The mean values of family size, consumers' wage, purchasers' gender, and educational level in the online environment are relatively higher than for offline consumers, demonstrating that consumers with better socio-economic status have a strong preference for online OM purchases, which is confirmed by Wang et al. [51]. In particular, the mean value of age in the online context is relatively smaller for offline customers, supporting the idea of Pham et al. [52] noting that young consumers are dominated by the online purchasing channel of organic foods. From the perspective of product attributes, the mean value of online external attributes surpasses that of offline attributes, while the former is smaller than the latter concerning the internal attributes of OM products.

This indicates that the influencing weight of external and internal attributes varies across channels. Particularly, we can notice that the mean value of online marketing resource intensity is larger than the corresponding value of the offline channel, showing the importance of an online marketing strategy for enterprises [18].

## 5. Empirical Results

Reports on the coefficient estimates of basic hedonic models are listed in Table 3 (i.e., estimates Equations (2) and (3)), and in both model specifications, we control for the fixed effect at the level of purchasing regions and time. Overall, the coefficients of both models are significant at the statistical level of 5%. In particular, the value of R-square regarding the hedonic regression model shows that the independent variables explain more than 90% of the variation in the ESG rating index. Therefore, the performance of the regression results is satisfactory.

**Table 3.** The hedonic regression results of the basic model.

| Variables | The Whole Sample | m = 1 | m = 0 |
|---|---|---|---|
| Fs | 0.417 *** (0.025) | 0.410 *** (0.048) | 0.403 *** (0.029) |
| w | 0.150 *** (0.083) | 0.118 *** (0.015) | 0.106 *** (0.011) |
| Gender | 0.390 ** (0.035) | 0.315 *** (0.059) | 0.203 *** (0.044) |
| Age | 0.018 *** (0.029) | 0.015 *** (0.058) | 0.012 *** (0.038) |
| Edu | 0.278 *** (0.035) | 0.242 *** (0.053) | 0.217 ***(0.030) |
| Pack | 0.483 *** (0.038) | 0.442 *** (0.069) | 0.239 *** (0.046) |
| Vers | 0.397 *** (0.074) | 0.224 *** (0.012) | 0.164 *** (0.095) |
| Fla | 0.325 *** (0.015) | 0.358 *** (0.016) | 0.368 *** (0.018) |
| Func | 0.276 *** (0.007) | 0.274 *** (0.011) | 0.279 *** (0.096) |
| Fat | 0.149 *** (0.035) | 0.116 *** (0.058) | 0.107 *** (0.034) |
| Sugar | 0.121 *** (0.024) | 0.107 *** (0.052) | 0.149 *** (0.033) |
| Probio | 0.168 *** (0.054) | 0.153 *** (0.086) | 0.162 * (0.025) |
| Kids | 0.117 *** (0.056) | 0.106 * (0.034) | 0.175 ** (0.039) |
| R-square | 0.962 | 0.932 | 0.915 |

Note: +Variables are in logarithmic form. The numbers in brackets are t-values. *** $p < 0.01$, ** $p < 0.05$, and * $p < 0.1$.

From this table, the coefficients of CSC variables show that the ESG rating index is positively associated with the consumers' socio-economic status, verifying the hypothesis H1. This is consistent with the studies from other scholars [31,53], suggesting that consumers' demographics, moderated by their awareness of CSR, have greatly affected the evaluation of CSR. Especially by comparing the coefficients of consumers, we find that the family size, purchasing gender, and educational level of consumers have a larger influence on the ESG performance of OM enterprises relative to purchasers' age, which is in line with the fact that female consumers with a higher educational level and income support the ESG management of enterprises [11]. Note that the positive coefficient of purchasers' age demonstrates that experienced consumers tend to focus on the ethical issue of enterprises [12,31].

From the perspective of product attributes, the corresponding positive coefficients demonstrate that heterogeneous product attributes significantly positively impact the ESG performance of enterprises, verifying hypothesis H1. This supports the ideas of Siegel and Vitaliano [54], further pointing out that the multi-product category is able to link the product itself into the ESG issue of enterprises by revealing the quality of experience products. This is explained by Tian et al. [31], who explained that enterprises focusing on the management of CSR are interpreted by the difference in consumers responses to CSR amid the multifarious product characteristics. In addition, we also provide evidence that the coefficients of external product attributes are relatively larger for OM products, which is supported by Jones et al. [55]. Jones et al. [56] further note that packaging can reveal product information concerning ESG commitment, implying that enterprises should pay more attention to the design of external packaging in order to enhance consumers' CSR perception and brand loyalty.

Moreover, the second and third columns of Table 3 show that the ESG performance of OM enterprises varies across channels. For instance, we find that the coefficients of online consumers' demographics are similar to those of offline consumers, which demonstrates that women with higher education levels and incomes have paid more attention to the ESG issue of enterprises [31]. Moreover, by comparing the coefficients of product attributes, we can notice that the external product attributes have a larger effect on the ESG effect in the online environment, while the internal product attributes significantly influence the ESG effect in the external product features. This implies that enterprises ought to implement the product differentiation strategy in the management of their ESG evaluation system in order to adapt to market demand from different customer segments.

To examine the robustness of the baseline regression, we adopt alternative measures of the ESG rating index: ESG data from the Wind Data Service Platform. Here, based on the thoughts of Fang et al. [3], we construct the ESG_W variable based on the following steps. Firstly, we select the ESG rating index of milk enterprises from the Wind Data Service Platform via Python and divide this research sample into 150 groups. Secondly, we assign a specific value of 1–100 based on the ESG rating. Especially, the explanatory variable ESG_W equals zero if the Wind-ESG evaluation rate is CCC, CC, C, D, and E, and one otherwise. Note that we control the fixed effects of purchasing time and location in this section. Table 4 reports the robustness check for the baseline regression. From this table, we can see that the coefficients of consumers' socio-economic features and product attributes are significant at the statistical level of 5%. In addition, by comparing the regression results of Tables 3 and 4, we find that the regression results of the baseline model are robust.

**Table 4.** The robustness check of the basic model.

| Variables | The Whole Sample | m = 1 | m = 0 |
|---|---|---|---|
| Fs | 0.964 *** (0.026) | 0.828 *** (0.031) | 0.630 *** (0.052) |
| w | 0.423 *** (0.083) | 0.412 *** (0.018) | 0.399 *** (0.013) |
| Gender | 0.444 *** (0.042) | 0.428 *** (0.053) | 0.423 *** (0.070) |
| Age | 0.016 *** (0.035) | 0.014 *** (0.057) | 0.013 *** (0.044) |
| Edu | 0.429 *** (0.042) | 0.420 *** (0.069) | 0.415 ***(0.053) |
| Pack | 0.375 *** (0.046) | 0.367 *** (0.056) | 0.342 *** (0.069) |
| Vers | 0.296 *** (0.083) | 0.217 *** (0.013) | 0.187 *** (0.011) |
| Fla | 0.360 *** (0.012) | 0.323 *** (0.017) | 0.433 *** (0.015) |
| Func | 0.238 *** (0.080) | 0.234 *** (0.010) | 0.249 *** (0.013) |
| Fat | 0.167 *** (0.042) | 0.134 *** (0.070) | 0.170 *** (0.053) |
| Sugar | 0.140 *** (0.043) | 0.130 *** (0.066) | 0.173 *** (0.042) |
| Probio | 0.189 *** (0.062) | 0.140 *** (0.010) | 0.189 *** (0.079) |
| Kids | 0.247 *** (0.067) | 0.230 *** (0.011) | 0.248 ** (0.085) |
| R-square | 0.945 | 0.946 | 0.925 |

Note: In this Table, we control for the fixed effects of purchasing time and region. The numbers in brackets are t-values. *** $p < 0.01$, and ** $p < 0.05$.

Table 5 displays the event analysis results of Equation (4). The overall estimated coefficients are positive at the 1% significance level, verifying the hypothesis H1. This shows that the regression results are robust. In addition, by comparing the coefficients of the second and third columns, we find that the latter has higher fitting coefficients compared with the former, which implies that the policy effect of FPI potentially enhances the ESG impact of consumers' socio-economic status and product attributes, which is evidenced by Sandberg et al. [1]. In particular, we also notice that the coefficients of product attributes show a more significant positive trend relative to the consumers' socio-economic status, which Park and Li [57] explained as the widespread application of Blockchain Technology in sustainable supply systems. This implies that OM enterprises should pay more attention to the production of differentiated products in the establishment of the ESG evaluation system. Note that the change in the coefficients of the family size variable (*fs*) is maximal compared with the other consumers' socio-economic variables (from 0.353 to 0.392), which

manifests that family size plays a crucial role in the construction of the ESG evaluation system for OM enterprises.

**Table 5.** The dynamic analysis of the basic model.

| Variables | Post = 0 | Post = 1 |
|---|---|---|
| Fs | 0.353 (0.027) | 0.392 ** (0.023) |
| w | 0.280 *** (0.013) | 0.294 *** (0.015) |
| Gender | 0.351 *** (0.043) | 0.373 *** (0.035) |
| Age | 0.034 (0.038) | 0.039 *** (0.033) |
| Edu | 0.341 *** (0.045) | 0.351 *** (0.037) |
| Pack | 0.317 (0.047) | 0.384 *** (0.038) |
| Vers | 0.214 (0.067) | 0.257 *** (0.085) |
| Fla | 0.337 *** (0.018) | 0.389 *** (0.068) |
| Func | 0.147 (0.083) | 0.217 *** (0.060) |
| Fat | 0.111 *** (0.043) | 0.198 *** (0.035) |
| Sugar | 0.095 *** (0.039) | 0.169 *** (0.036) |
| Probio | 0.135 *** (0.067) | 0.217 *** (0.051) |
| Kids | 0.239 * (0.071) | 0.302 *** (0.060) |
| R-square | 0.896 | 0.897 |

Note: In this table, we control for the fixed effects of purchasing time and region. The numbers in brackets are t-values. *** $p < 0.01$, ** $p < 0.05$, and * $p < 0.1$.

Figure 2 displays the dynamic analysis results in a multi-channel context. From this figure, we find that the lines representing the dynamic estimated coefficients are steeper in the online environment relative to the traditional offline channel, which signifies that the policy effect of the ESG evaluation system regarding the consumers' socio-economic status and product attributes is more significant in the online channel. The previous scholar reveals the dynamic effect of the ESG rating index in the tourism industry [58]. We further provide evidence that consumers and enterprises pay more attention to the online dynamic effect of the ESG rating index, implying that the OM enterprises ought to develop the ESG evaluation system based on the online environment.

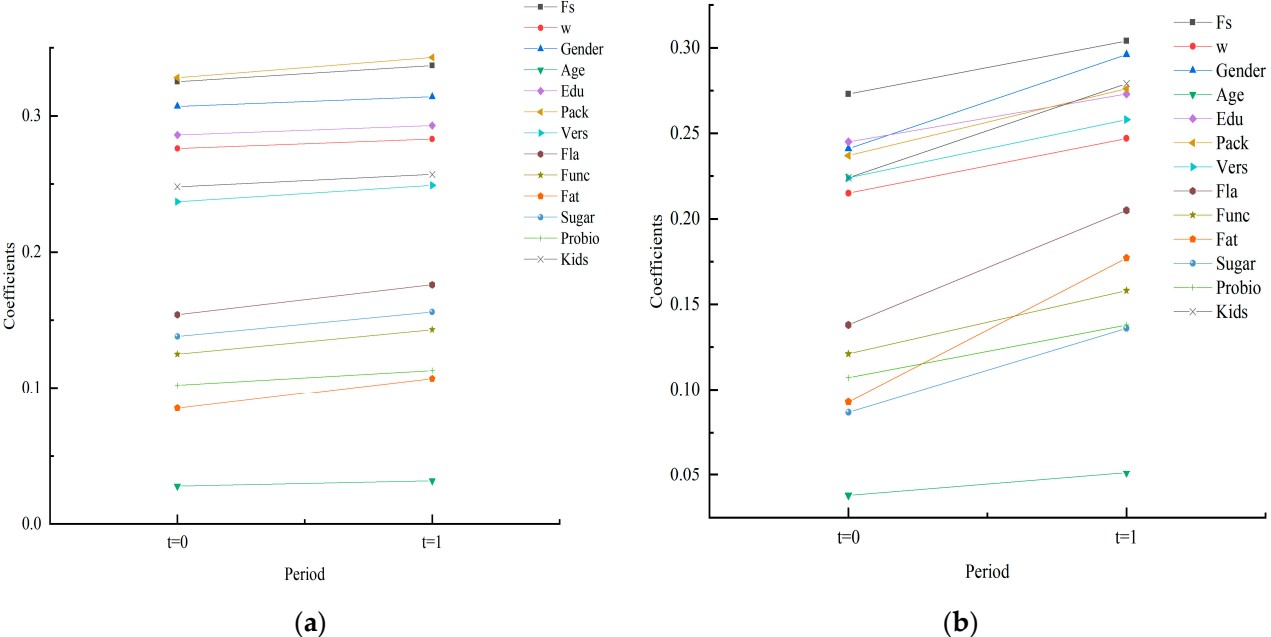

**Figure 2.** The dynamic analysis of the based model in the multi-channel context. (**a**) The traditional offline channel (m = 0) and (**b**) the online channel (m = 1).

In addition, we can notice that the coefficients of gender tend to sharply increase from 0.215 to 0.275 in the online environment, which shows that female consumers are more

keen on acquiring the OM product via online shopping relative to male consumers [48]. This means that OM retailers ought to take the time to listen to the voices of the female customers in the online market. Note that, in the online environment, the changes in the coefficients of flavor and fat variables are maximal compared with other product attributes, demonstrating that the online consumers are more concerned about the internal quality of OM products, which is confirmed by Park and Li [57], implying that OM producers should pay more attention to the internal operation management of OM food quality in the construction of the online ESG evaluation system.

Table 6 presents the estimated sub-ESG rating index for the OM product. From this Table, the coefficients of consumers' socio-economic features and product attributes are significant at the 5% significance level. We can also notice that the above explanatory variables are positively associated with the sub-ESG performance of OM enterprises, verifying hypothesis H1. This is supported by Tian et al. [31]. Moreover, from the perspective of demographic impacts, we find that other consumers' socio-economic features have a more significant influence on the sub-ESG performance of OM products relative to the purchasers' age, which is in line with the work of Youn and Kim [11] indicating that female customers with higher educational levels and income tend to focus on the CSR issue of enterprises. In particular, the coefficients of the family size variable (*fs*) are maximal compared with other consumers' socio-economic status. This is explained by Chu et al. [59] as the size loyalty effect of consumers' purchasing behavior, which is defined as the sensitivity of the change in household size to the amount of goods purchased. Regarding the variable of product attribute, the corresponding positive coefficients demonstrate that the product attributes are positively linked to the sub-ESG performance of OM enterprises. Note that, by comparing the coefficients of product attributes, we can notice that the internal product attributes have a larger effect on the sub-ESG performance of OM enterprises, verifying the hypothesis H2. This is supported by Sandberg et al. [1], who imply that food enterprises ought to focus on the improvement of product quality in the process of ESG management. Moreover, by comparing the coefficients of three sub-ESG rating indexes, we can witness that the corresponding value of E_score is maximal, followed by S_score and G_score, which is in line with the work of Yu et al. [60] noting that Environmental CSR can improve the corporate image of organic products moderated by consumers' trust. This implies that enterprises ought to focus on the environmental issue of OM production and enhance the corresponding management of this product.

**Table 6.** Mechanism analysis of the ESG's performance.

| Variables | E_Score | S_Score | G_Score |
|---|---|---|---|
| Fs | 0.577 *** (0.047) | 0.566 *** (0.048) | 0.552 *** (0.043) |
| w | 0.378 *** (0.083) | 0.354 *** (0.021) | 0.322 *** (0.017) |
| Gender | 0.453 *** (0.080) | 0.437 *** (0.080) | 0.468 *** (0.080) |
| Age | 0.073 *** (0.067) | 0.072 *** (0.065) | 0.069 *** (0.064) |
| Edu | 0.388 *** (0.079) | 0.316 *** (0.080) | 0.217 ***(0.081) |
| Pack | 0.289 ** (0.088) | 0.247 *** (0.089) | 0.225 *** (0.083) |
| Vers | 0.196 *** (0.074) | 0.178 *** (0.086) | 0.149 *** (0.088) |
| Fla | 0.388 *** (0.017) | 0.339 *** (0.015) | 0.315 *** (0.010) |
| Func | 0.320 *** (0.014) | 0.315 *** (0.009) | 0.310 *** (0.015) |
| Fat | 0.349 *** (0.081) | 0.325 *** (0.080) | 0.319 *** (0.067) |
| Sugar | 0.454 *** (0.017) | 0.417 *** (0.018) | 0.387 *** (0.012) |
| Probio | 0.583 *** (0.078) | 0.515 *** (0.011) | 0.439 *** (0.014) |
| Kids | 0.231 *** (0.013) | 0.215 *** (0.012) | 0.203 ** (0.012) |
| MRI * fs | 0.354 ** (0.017) | 0.324 *** (0.013) | 0.308 * (0.007) |
| MRI * w | 0.397 *** (0.077) | 0.254 *** (0.072) | 0.118 ** (0.062) |
| MRI * gender | 0.309 *** (0.029) | 0.237 *** (0.028) | 0.190 *** (0.030) |
| MRI * age | 0.229 *** (0.031) | 0.218 (0.004) | 0.212 *** (0.017) |
| MRI * edu | 0.378 *** (0.028) | 0.341 *** (0.023) | 0.315 *** (0.012) |

**Table 6.** *Cont.*

| Variables | E_Score | S_Score | G_Score |
|---|---|---|---|
| MRI * pack | 0.013 (0.032) | 0.062 *** (0.027) | 0.057 *** (0.033) |
| MRI * vers | 0.026 ** (0.051) | 0.026 *** (0.034) | 0.024 *** (0.045) |
| MRI * fla | 0.179 *** (0.056) | 0.128 *** (0.057) | 0.107 *** (0.061) |
| MRI * func | 0.136 *** (0.053) | 0.128 *** (0.055) | 0.112 (0.064) |
| MRI * fat | 0.150 *** (0.029) | 0.137 *** (0.027) | 0.114 *** (0.030) |
| MRI * sugar | 0.166 *** (0.029) | 0.115 *** (0.018) | 0.102 ** (0.045) |
| MRI * probio | 0.184 *** (0.039) | 0.154 *** (0.032) | 0.128 *** (0.043) |
| MRI * kids | 0.125 ** (0.047) | 0.113 *** (0.046) | 0.105 *** (0.041) |
| R-square | 0.897 | 0.902 | 0.912 |

Note: The numbers in brackets are t-values. *** $p < 0.01$, ** $p < 0.05$, and * $p < 0.1$.

The coefficients of interaction terms regarding the MRI variable show that marketing resource intensity plays a positive moderating role in the ESG management of OM enterprises, verifying the hypothesis H1. This is confirmed by Ullah [18], who notes that the brand value can be recognised multi-channel consumers via the reasonable application of marketing resources. Particularly, by comparing the coefficients of the MRI interaction term, we can notice that the coefficients of consumers' socio-economic status are relatively larger for the ESG performance of OM enterprises. This demonstrates that consumers pay more attention to the marketing strategy of OM enterprises because the marketing resources are capable of revealing the product quality to some extent [61]. Dangelico [62] further figures out that the MRI tool can link the sustainable purchase of green products into the ESG performance of enterprises, which is the same with the industry of OM products. Therefore, enterprises ought to reasonably employ marketing resources to maintain the user stickiness and market loyalty of consumers.

## 6. Discussion, Implications, and Limitations

This study aims to investigate how consumers' socio-economic features and product attributes affect the ESG performance of OM products.

We demonstrate that consumer socio-economic characteristics and product attributes, moderated by the MRI, have a positive impact on the ESG rating index. Moreover, the environmental disclosure index plays a more positive role compared to the social and governance disclosure indexes. The findings of the study can intuitively guide the determination of corporate marketing strategies and the establishment of competitiveness improvement measures, and they can contribute to the construction of an ESG comprehensive evaluation system and the implementation of ESG tools.

The marketing implications of these clear-cut findings are listed as follows. Firstly, marketing to consumer groups with a high socio-economic status can be chosen, for instance, in the organic milk market, where the econometric analysis shows that the online purchasing environment is superior to the offline environment. Secondly, the introduction of the MRI tool can better regulate the corporate marketing process. The findings also suggest that diversified marketing tools and differentiated product design can improve the company's competitiveness and achieve long-term profitability. Thirdly, the study of the sub-ESG disclosure index illustrates the important impact of environmental factors, and green product design and transparent environmental information disclosure can effectively contribute to the improvement of the company's brand image.

Companies can also increase their competitiveness by introducing MRI tools and differentiating their product designs, which can lead to long-term profitability. The study of the sub-ESG evaluation index demonstrates the significance of environmental variables and implies that green product design and open disclosure of environmental information could potentially improve the company's brand image.

Additionally, the regression results of the ESG rating index enable us to claim that FPI has an impact on OM producers' ESG from a policy standpoint. The sub-ESG impact of OM is significantly moderated by MRI tools. ESG and sub-ESG rating indexes are

significantly influenced by the product's intrinsic characteristics. The coefficient of the household size variable shows that household size has a significant impact on the ESG or sub-ESG performance of OM.

The previous literature has widely employed the theories of EPB and ECB to account for the CSR impact of consumers' socio-economic status and product attributes [2,11,16,17]. Marketing studies on the ESG and sub-ESG effects of the above explanatory variables, especially for the market of OM products, are limited in the business and economic literature [32]. Even though some scholars have explored the ethical purchasing motivation for OM products from the perspective of the consumers' socio-economic status [24], the research on the ESG effect of product attributes is seldom discussed in the prior literature, which will provide a new angle of view for the understanding of the ESG performance of ethical enterprises.

Therefore, the impact of consumers' socio-economic characteristics on ESG and sub-ESG implies that the construction of ESG evaluation systems should be based on market research on consumer demographics. This is why corporate marketing strategies should be built with a human face and in line with the ethical requirements of consumers. Moreover, OM companies should improve the design of differentiated products to improve the social performance of feasible green products (such as OM products), as well as establish a transparent supply chain system that enables customers to trace the origin and production process of their products. This is indicated by a positive ESG coefficient for product attributes. Boost product credibility among consumers.

In addition, the policy analysis of ESG performance suggests that OM companies should focus on producing differentiated food products in the context of FPI policy. Offering a wide range of organic milk products, such as whole, low-fat, and skim milk, to meet the needs and preferences of different consumers.

More importantly, the positive moderating effect of the MRI implies that the company's ESG performance cannot be separated from the implementation of a diversified marketing strategy, and the social impact of the ESG evaluation system should be continuously amplified. In the real management process, producers should pay more attention to the construction of brand image, which can provide product quality information related to OM and increase consumer awareness and understanding of this milk product.

The coefficients of the sub-ESG rating index demonstrate that producers should consider the environmental impact of saving energy, reducing the carbon footprint, and promoting sustainable packaging materials, as well as their active participation in agricultural sustainability projects that include the cultivation of organic farming and biodiversity conservation.

As with any study, this research is not without limitations. Firstly, the current study is limited by the small number of cases analysed and the lack of case studies of ESG performance across different cultural contexts and industries. In addition, the ESG rating index is potentially affected by consumers' psychology [63–65], which is not discussed in this paper due to the limitation of the dataset structure. Overall, due to the importance of corporate social responsibility [66–68], this study checks the two hypotheses proposed, explores and demonstrates the influencing mechanisms of the ESG rating index in economic and business activities, and provides a fresh perspective based on traditional research. The formation of sub-ESG performance of accessible green foods (e.g., OM products) is further analysed, providing a strong basis for the construction of an ESG evaluation system for the OM industry.

Based on the above analysis, the future direction of related research includes two main points. On the one hand, the analysis of different types of companies by expanding case sources will explore the generalisability and practical application of ESG evaluation systems. On the other hand, consumer psychology is introduced as an explanatory variable, and the ESG performance of OM producers will be further analysed in conjunction with market survey data to reveal the ESG-psychosocial impact of OM products.

**Author Contributions:** Conceptualisation; data collection; formal analysis; manuscript writing—original draft, J.H.; literature search; writing—original draft, review, and editing, Y.W.; conceptualisation, formal analysis, supervision, manuscript—review and editing, M.D. All authors have read and agreed to the published version of the manuscript.

**Funding:** This study was supported by the Beijing Commercial Development Research Center (NO. JD-YB-2022-055) and the Key Program of the National Social Science Foundation of China (NO. 21ATJ007).

**Institutional Review Board Statement:** Not applicable.

**Informed Consent Statement:** Not applicable.

**Data Availability Statement:** The data that support the findings of this study are available from CTR Market Research Co. Ltd., but restrictions apply to the availability of these data, which were used under licence for the current study and are therefore not publicly available. Data are, however, available from the authors upon reasonable request and with permission of CTR Market Research Co., Ltd.

**Acknowledgments:** We want to thank Qingjie Zhou from Beijing Technology and Business University for the helpful discussions, comments and supports, and thank for the assistance of the proofreader—Wennanxiang Wang from Huaqiao University.

**Conflicts of Interest:** The authors declare no conflict of interest.

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
