# Peer review of "Friend or Foe: How Do Consumers and Producers Affect the ESG Rating Index? Evidence from China’s Market of Organic Milk"

_sustainability, doi:10.3390/su151410819_

Round 1
Reviewer 1 Report
Dear authors,
Thanks for this paper. The study aims to demonstrate the consumers’ socioeconomic features and product attributes positively affect the ESG performance of enterprises, moderated by marketing resource intensity. This study is different by advancing the hedonic model of ESG and sub-ESG rating index combined with the ESG rating index from Bloomberg Reports and Wind Data Service Platform dataset and by including variable of marketing resource intensity (MRI) in order to reveal the influencing mechanism of ESG impact.
Regarding literature and theoretical discussion, the paper lacks quality in terms of theoretical discussion and arguments even though they defined two hypotheses separately. They superficially discuss the constructs of ethical product behavior (EPB) and ethical consumption behavior (ECB). Regarding the method, the authors employed the hedonic regression model to analyze the hypotheses and they explained the method in detail. They explained the findings and the discussion section in a clear way, however, the connection with the theoretical discussion is weak and must be improved.
Author Response
Responses to Reviewers’ Comments
Dear reviewer,
We are very grateful to the reviewers for their professional comments on our articles. These comments were very valuable and helped us to improve the quality of the manuscript by revising any problems in the article. Based on the suggestions of the comments, we have refined the framework structure of the article to present the research objectives, research questions, research innovations and research conclusions as clearly as possible. The linkage between the theoretical framework and the subsequent empirical evidence in the second subsection has been strengthened to make the argument more convincing. The data analysis is relatively well developed, but given the limitations of the data, we have also proposed the expected research direction of adding more cases and introducing consumer psychology. In the revised version of the document, changes to the manuscript are highlighted in green. Below are detailed responses to specific questions from various reviewers:
- Regarding literature and theoretical discussion, the paper lacks quality in terms of theoretical discussion and arguments even though they defined two hypotheses separately. They superficially discuss the constructs of ethical product behavior (EPB) and ethical consumption behavior (ECB). Regarding the method, the authors employed the hedonic regression model to analyze the hypotheses and they explained the method in detail. They explained the findings and the discussion section in a clear way, however, the connection with the theoretical discussion is weak and must be improved.
Answer:
Thanks for your advice. We have reorganized the link between the theoretical framework and the empirical analysis that follows. The theoretical part is based on the ethical product behavior (EPB) and ethical consumption behavior (ECB) theories, and it can be determined that the EGS rating index is effective in responding to the performance of organic milk products on environmental, social and governance issues. In contrast, during the practical argumentation in the later subsections, the hedonic regression model is first extended to identify the influencing variables of the ESG rating index as the demographic characteristics of consumers and product attributes.
These variables are in line with the ECB theory that "consumers consciously choose the products they buy based on their moral beliefs". The further development of an offline segmentation model and the decomposition of sub-ESG indices (environmental, social and governance) is in line with the EPB theory that "CSR and production activities are based on meeting consumer demands".
Ultimately, the results of the data analysis provide a visual indication of whether the assumptions of the theoretical part are true and whether they are supported by the data.
According to your suggestion, we have rewritten the theoretical framework and research hypothesis as follows:
“For organic milk (OM) products, the previous literature mainly discuss the ESG impact of consumers’ purchasing behavior from the following aspects: (i) Environment: carbon emissions and energy consumption can hurt the environment; organic milk production requires a large amount of water, which is prone to waste and pollution; (ii) Society: dairy companies need to ensure product quality and safety, including regulation and control from supply chain management to production processes; (iii) Corporate governance: the disclosure of relevant information by dairy companies including financial information, corporate governance structures and transparency of decision-making processes needs to be addressed at present.
Research on CSR offers details on the constructs of ethical product behavior (EPB) and ethical consumption behavior (ECB), corresponding to the demand and supply theory of CSR [19,20]. Sen [63] and Mohr et al. [67] figure out that the EPB and ECB potentially increase the sustainability of consumption and production, thereby significantly impacting the CSR performance of consumers and producers. Thus, some scholars define the above conceptions before analyzing the ESG impact of socioeconomic status and product attributes. Regarding the former, the theory of ECB is able to describe the moral effect of consumption behavior [21], which is by Auger et al. [22] called as ethical consumerism. ECB is by Crane and Matten [23] defined as “the conscious and deliberate purchasing decision based on moral belief and values”, which contributes to the better understanding of the consumers’ sustainable purchasing behavior of OM product [24,25].
In fact, government agencies can use the ESG evaluation system to ascertain the social performance of the industry and then create a more reasonable and consumer-friendly food production licensing system, thereby accomplishing macro-level market regulation[19]. Corporate managers can develop their upcoming competitive strategies and marketing tools based on the evaluation's results to increase their competitiveness and get long-term advantages [21]. According to the ESG indicators, consumers can choose goods that match their ethical standards [21].
Based on the theory of ECB, consumers’ purchasing behavior is determined by ethical obligation and CSR [26], which is by Al-Haddad et al. [27] explained as the moderating effect of consumer participation through social media. Concerning the latter, the theory of EPB links CSR into the enterprises’ production behavior, which is defined as “the overall process of ethical product design and production in order to meet the consumers’ demand for ethically sustainable products” [19]. The above theories are closely associated with the enterprises’ non-financial investment activities. This can be measured by the ESG rating index, which by Clementino and Perkins [28] is defined as “the expert’ s evaluation of a company based on a comparative assessment of their quality, standard or performance on environmental, social or governance issues”. Therefore, the theories of ECB and EPB can be successfully applied into the econometric analysis of ESG performance regarding OM product.
From the perspective of business management, ESG non-financial investment activities have gradually transformed into their competitive strategy [29], which implies that enterprises are responsible for public interests while seeking for economic benefits [30]. Thus, these activities contribute to the establishment of the enterprises’ ethical image and formation of user stickiness of consumers [2,10]. From the perspective of consumers’ socioeconomic status, Carrigan et al. [12] and Tian et al. [31] find that experienced consumers with higher socioeconomic features (e.g., larger family size, female consumers with higher income and educational level, etc) tend to focus on the ethical issue of OM enterprises. Consumers trusting in these enterprises have a willingness to pay for the corresponding branded products [32]. This is by Ullah [18] explained as the moderating role of marketing resource intensity (MRI), which is defined as the extent of enterprises’ investment in marketing activities including advertising, consumer awareness and consumer service [33]. Wagner [34] argues that advertising activities form the long-term cooperation relationship between buyers and sellers. De Boer [35] further contends that the maintenance of consumer awareness and consumer service can enhance the market communication by sending the information of enterprises’ sustainable endeavor.
From the perspective of producers, the EPB of producers is closely associated with their ESG performance, which can reveal the product quality and affect the consumers’ purchasing attitude [66]. Thus, product attributes have attracted increasing attention from consumers [22], which can be explained as the fact that product attributes are closely linked to the corporate association-the reflection of ESG [36]. Through the moderating power of MRI, the positive information of product attributes can be received by the ethical consumers, thereby generating differentiated product advantages for OM product and resulting in the willingness-to-pay (WTP) for this product [18].
Based on EPB and ECB theory, it can be determined that the EGS rating index is a valid response to the performance of organic milk products on environmental, social and governance issues. During the practical argumentation in the subsequent subsections, the hedonic regression model is first extended to identify the influencing variables of the ESG rating index as the demographic characteristics of consumers and product attributes. These variables are in line with the ECB theory that "consumers consciously choose the products they buy based on their moral beliefs". The further development of an offline segmentation model and the decomposition of sub-ESG indices (environmental, social and governance) is in line with the EPB theory that "CSR and production activities are based on meeting consumer demands". The results of the data analysis will ultimately show whether the assumptions of the theoretical part are true and whether they are supported by the dataset.
This study aims to examine how consumers' socio-economic characteristics and product attributes affect the ESG performance of OM products, while market research based on consumer demographics can further improve the construction of ESG evaluation systems. Only by understanding the influence mechanisms of the relevant variables can we find effective strategies and diversified marketing approaches to improve the competitiveness of companies.”
Thank you very much for your suggestions and patient guidance, and we have revised and improved the relevant issues. Looking forward to hearing from your response.
Thank you and beat regards.
Sincerely,
Jiangyuan Hou
School of International Trade and Economics, University of International Business and Economics, No. 10 Huixin East Road, Chaoyang District, Beijing, 100029, China
Yanping Wang
School of Economics, Henan University of Economics and Law, No. 108 Jinshui East Road, Zhengzhou, He-nan, 450011, China
Mingyue Du
School of Economics, Beijing Technology and Business University, No. 11 Fucheng Road, Haidian District, Beijing, 100048, China

Reviewer 2 Report
Thank you for the opportunity to review this manuscript. The authors propose a hedonic model to evaluate the ESG and sub-ESG impact of organic dairy products on the performance of enterprises in China.
Some suggestions to consider when reviewing this paper:
- I recommend improving the "abstract" by clearly mentioning the purpose of the research, the analysis method (hedonic regression model) and the contribution made,
- I recommend the introduction of a short paragraph related to the meaning of ESG. Present the investment philosophy, the need to implement socio-economic responsibility standards in evaluating and maximizing the profitability/performance of Chinese enterprises.
- Regarding the organic milk (OM) market in China, present the evolution of sales and profits in the period 2005-2020. In this context, what were the effects generated by the 2019-2020 COVID-19 pandemic.
- In the "introduction" the decision-making problem of your research is missing. What is the question that your study answers by developing this hedonic model?
- What are the main environmental, social and governance issues facing dairy enterprises in China?
- Present clearly and concisely what is the purpose of your research. Delineate the secondary objectives and the way to achieve them.
- The contribution could be improved by clearly highlighting the scientific and practical contribution obtained after evaluating the ESG impact on the company's performance and after using the moderating variable - marketing resources. Introduce the usefulness of the study for government institutions, managers and consumers of dairy products.
- The "Literary Magazine" section can be improved. Present aspects related to the legislative framework, the ethical behavior of the consumer, the moderating effect of social networks on the behavior of the consumer, the ESG rating index, the ethical image of the company, the socio-economic status of consumers, etc. creating a correlation and a staging between them, depending on research objectives/hypotheses. Enter several bibliographic sources to support these significant influences
- Section 4- "Data" - The data collection and processing method is very succinctly presented - Line 208-211. What are the limiting conditions?
- In the "Discussion" section, you can create a link between the results obtained, the problems found in practice and the literature review, you can formulate proposals for government organizations/managers involved in the Chinese dairy market
- Outline the scientific and practical implications of the “Discussion” section and add future directions.
Author Response
Responses to Reviewers’ Comments
Dear reviewer,
We are very grateful to the reviewers for their professional comments on our articles. These comments were very valuable and helped us to improve the quality of the manuscript by revising any problems in the article. Based on the suggestions of the comments, we have refined the framework structure of the article to present the research objectives, research questions, research innovations and research conclusions as clearly as possible. The linkage between the theoretical framework and the subsequent empirical evidence in the second subsection has been strengthened to make the argument more convincing. The data analysis is relatively well developed, but given the limitations of the data, we have also proposed the expected research direction of adding more cases and introducing consumer psychology. In the revised version of the document, changes to the manuscript are highlighted in green. Below are detailed responses to specific questions from various reviewers:
- 1. I recommend improving the "abstract" by clearly mentioning the purpose of the research, the analysis method (hedonic regression model) and the contribution made.
Answer:
We have revised the abstract to clearly state the purpose of the study. According to your suggestion, we have rewritten the abstract:
“This study attempts to investigate how product attributes and consumer socio-economic status affect OM product ESG performance. There aren't many pertinent studies on the interaction between the EGS evaluation system and organic milk products. Thus, this paper develops a targeted hedonic-regression model based on the EPB and ECB theories to investigate the ESG and sub-ESG impact of the above attributes based on the consumption dataset from the Kantar Worldpanel. We also introduce the moderating variable of marketing resource intensity (MRI) to check the influencing mechanism of sub-ESG rating index. This study demonstrates that the intensity of marketing resources has the moderating effect on the ESG performance of organic milk producers. Moreover, we also evidence that the household’ s family size and internal product attributes have a more significant impact on the ESG rating index. In addition, compared to social and governance information, the environmental information disclosure index has a more significant influence on the consumers’ socioeconomic status and product attributes compared with social and governance rating index. This study not only contributes to deepening the understanding of ESG evaluation system, but also provide a scientific guidance for the producers’ ESG strategy. Therefore, to increase the competitive advantage of OM enterprises, they should endeavor to construct the comprehensive evaluation system of entertainment social governance rating index and engage in the environmental investment activities.”
- I recommend the introduction of a short paragraph related to the meaning of ESG. Present the investment philosophy, the need to implement socio-economic responsibility standards in evaluating and maximizing the profitability/performance of Chinese enterprises.
Answer:
In the second sub-section, we name the actual definition of ESG, specifically:
“ESG issues including three sub-ESG rating index- environmental, social, and corporate governance refer to the assessment of a company's performance and impact in these areas. Analysing the impact of ESG evaluation systems can improve a company's core competencies and identify areas for optimisation, which can ultimately generate long-term value, reduce risk and have a positive impact on society and the environment.”
- Regarding the organic milk (OM) market in China, present the evolution of sales and profits in the period 2005-2020. In this context, what were the effects generated by the 2019-2020 COVID-19 pandemic.
Answer:
We added Table 2 to the data of the fourth subsection, explaining the actual milk production in China from 2005-2020. The results showed that Organic milk production is on an upward trend, with a more significant rise during the 2019-2020 COVID-19 pandemic. Due to the long-term home-based office status and the awakening of health and hygiene awareness [68], organic milk industry has achieved a 14.7% increase in sales in China, accounting for 29% of consumption expenditure at home but only 10% of consumption expenditure away from home.
- Xu, J.; Wang, J; Li, C. Impact of Consumer Health Awareness on Dairy Product Purchase Behavior during the COVID-19 Pandemic. 2022, 14, 314.
- In the "introduction" the decision-making problem of your research is missing. What is the question that your study answers by developing this hedonic model?
Answer:
We explicitly address the research question in the first subsection. (i) How do consumers’ socioeconomic status and product attributes affect the ESG and sub-ESG performance of OM producers? (ii) How are the influencing mechanisms of ESG rating index regarding the above attributes? (iii) How do producers carry out the ESG strategy to improve their market competitiveness?
- What are the main environmental, social and governance issues facing dairy enterprises in China?
Answer:
In the second section, based with the actual situation of Chinese organic milk market, we clearly proposed the ESG (environmental, social and corporate governance) problems of Chinese dairy enterprises: (i) Environment: carbon emissions and energy consumption can hurt the environment; organic milk production requires a large amount of water, which is prone to waste and pollution; (ii) Society: dairy companies need to ensure product quality and safety, including regulation and control from supply chain management to production processes; (iii) Corporate governance: the disclosure of relevant information by dairy companies including financial information, corporate governance structures and transparency of decision-making processes.
- Present clearly and concisely what is the purpose of your research. Delineate the secondary objectives and the way to achieve them.
Answer:
The research purpose of this study is clearly stated in the revised introduction section. This aim is to investigate the ESG effect of consumers’ OPBs of OM. The following issues will be addressed in this paper: (i) How do consumers’ socioeconomic status and product attributes affect the ESG and sub-ESG performance of OM producers? (ii) How are the influencing mechanisms of ESG rating index regarding the above attributes? (iii) How do producers carry out the ESG strategy to improve their market competitiveness? Based on the theory of EPB and ECB, this research achieves these targets via the following path. Firstly, this study delves into the mechanisms by which consumers' socio-economic status and product attributes influence the ESG rating index, rather than simply analysing its impact on social responsibility performance. Secondly, the moderating variable of marketing resource intensity (MRI) is introduced into our econometric analysis in order to link the marketing tools into the ESG performance of OM producers. Thirdly, the hedonic analysis is conducted for the sub-ESG rating index, and the importance of environmental disclosure index is argued.
- The contribution could be improved by clearly highlighting the scientific and practical contribution obtained after evaluating the ESG impact on the company's performance and after using the moderating variable - marketing resources. Introduce the usefulness of the study for government institutions, managers, and consumers of dairy products.
Answer:
The research purpose of this study is clearly stated in the revised introduction section. This aim is to investigate the ESG effect of consumers’ OPBs of OM. The following issues will be addressed in this paper: (i) How do consumers’ socioeconomic status and product attributes affect the ESG and sub-ESG performance of OM producers? (ii) How are the influencing mechanisms of ESG rating index regarding the above attributes? (iii) How do producers carry out the ESG strategy to improve their market competitiveness? Based on the theory of EPB and ECB, this research achieves these targets via the following path. Firstly, this study delves into the mechanisms by which consumers' socio-economic status and product attributes influence the ESG rating index, rather than simply analysing its impact on social responsibility performance. Secondly, the moderating variable of marketing resource intensity (MRI) is introduced into our econometric analysis to link the marketing tools into the ESG performance of OM producers. Thirdly, the hedonic analysis is conducted for the sub-ESG rating index, and the importance of environmental disclosure index is argued.
- The "Literary Magazine" section can be improved. Present aspects related to the legislative framework, the ethical behavior of the consumer, the moderating effect of social networks on the behavior of the consumer, the ESG rating index, the ethical image of the company, the socio-economic status of consumers, etc. creating a correlation and a staging between them, depending on research objectives/hypotheses. Enter several bibliographic sources to support these significant influences.
Answer:
We modified the structure of the second section, first confirmed the significance of the research problem and the ESG evaluation system, then provided theoretical support for this research through the discussion of recent literature, and finally supplemented the connection between theory and example demonstration and confirmed that the hypothesis proposed in this study is reasonable and has research significance.
- Section 4- "Data" - The data collection and processing method is very succinctly presented - Line 208-211. What are the limiting conditions?
Answer:
Thanks to reviewer for the question. Firstly, we modified the structure of the second section based on your advice. We confirmed the significance of the research problem and the ESG evaluation system, then provided theoretical support for this research through the discussion of recent literature, and finally supplemented the connection between theory and example demonstration and confirmed that the hypothesis proposed in this study is reasonable and has research significance.
The limiting conditions are as follows:
As with any study, this research is not without limitations. Firstly, the current study is limited by the small number of cases analysed and the lack of case studies of ESG performance across different cultural contexts and industries. In addition, the ESG rating index is potentially affected by consumers’ psychology [63], which is not discussed in this paper due to the limitation of dataset structure. Overall, this study checks the two hypotheses proposed, explores and demonstrates the influencing mechanisms of ESG rating index in economic and business activities, and provides a fresh perspective based on traditional research. The formation of sub-ESG performance of accessible green foods (e.g., OM products) is further analysed, providing a strong basis for the construction of ESG evaluation system for the OM industry.
Based on the above analysis, the future direction of related research includes two main points. On the one hand, the analysis of different types of companies by expanding case sources will explore the generalisability and practical application of ESG evaluation systems. On the other hand, consumer psychology is introduced as an explanatory variable and ESG performance of OM producers will be further analysed in conjunction with market survey data to reveal the ESG-psychosocial impact of OM products.
- In the "Discussion" section, you can create a link between the results obtained, the problems found in practice and the literature review, you can formulate proposals for government organizations/managers involved in the Chinese dairy market.
Answer: Following the conclusions of this study, we modified the content framework of the last subsection and added relevant recommendations for enterprises. The specific content is:
Therefore, the impact of consumers' socio-economic characteristics on ESG and sub-ESG implies that the construction of ESG evaluation systems should be based on market research on consumer demographics. This is why corporate marketing strategies should be built with a human face and in line with the ethical requirements of consumers. Moreover, OM companies should improve the design of differentiated products to improve the social performance of feasible green products (such as OM products), as well as by establishing a transparent supply chain system that enables customers to trace the origin and production process of their products. This is indicated by a positive ESG coefficient for product attributes. Boost product credibility among consumers.
In addition, the policy analysis of ESG performance suggests that OM companies should focus on producing differentiated food products in the context of FPI policy. Offering a wide range of organic milk products, such as whole, low-fat and skim milk, to meet the needs and preferences of different consumers.
More importantly, the positive moderating effect of the MRI implies that the company's ESG performance cannot be separated from the implementation of a diversified marketing strategy and the social impact of the ESG evaluation system should be continuously amplified. In the real management process, producers should pay more attention to the construction of brand image, which can provide information product quality information related to OM and increase consumer awareness and understanding of this milk product.
The coefficients of sub-ESG rating index demonstrate that producers should consider the environmental impact of saving- energy, reducing the carbon footprint and promoting sustainable packaging materials as well as the active participation in agricultural sustainability projects containing including the cultivation of organic farming and biodiversity conservation.
As with any study, this research is not without limitations. Firstly, the current study is limited by the small number of cases analysed and the lack of case studies of ESG performance across different cultural contexts and industries. In addition, the ESG rating index is potentially affected by consumers’ psychology [63], which is not discussed in this paper due to the limitation of dataset structure. Overall, this study checks the two hypotheses proposed, explores and demonstrates the influencing mechanisms of ESG rating index in economic and business activities, and provides a fresh perspective based on traditional research. The formation of sub-ESG performance of accessible green foods (e.g. OM products) is further analysed, providing a strong basis for the construction of ESG evaluation system for the OM industry.
Based on the above analysis, the future direction of related research includes two main points. On the one hand, the analysis of different types of companies by expanding case sources will explore the generalisability and practical application of ESG evaluation systems. On the other hand, consumer psychology is introduced as an explanatory variable and ESG performance of OM producers will be further analysed in conjunction with market survey data to reveal the ESG-psychosocial impact of OM products.
- Outline the scientific and practical implications of the “Discussion” section and add future directions.
Answer:
Thanks for the reviewer’s advice. We have further revised the content of the manuscript for future directions, firstly by presenting the shortcomings of the current research and the future directions precisely to address the existing problems. On the one hand, the generalisability and practical application of ESG evaluation systems are explored by expanding case sources and analyzing different types of companies. On the other hand, consumer psychology is introduced as an influencing variable and ESG performance is further analysed in conjunction with market survey data to reveal the ESG-psychosocial impact of OM products. The details are shown as follows:
Scientific and practical implications:
We demonstrate that consumer socio-economic characteristics and product attributes, moderated by the MRI, have a positive impact on the ESG rating index. Moreover, The environmental disclosure index plays a more positive role compared to the social and governance disclosure index. The findings of the study can intuitively guide the determination of corporate marketing strategies and the establishment of competitiveness improvement measures, and contribute to the construction of ESG comprehensive evaluation system and the implementation of ESG tools. Marketing to consumer groups with a high socioeconomic status can be chosen, for instance, in the organic milk market, where the econometric analysis shows that the online purchasing environment is superior to the offline environment.
Companies can also increase their competitiveness by introducing MRI tools and differentiating their product designs, which can lead to long-term profitability. The study of sub-ESG evaluation index demonstrates the significance of environmental variables and implies that green product design and open disclosure of environmental information potentially improve the company's brand image.
Additionally, the regression results of ESG rating index enable us to claim that FPI has an impact on OM producers' ESG from a policy standpoint. The sub-ESG impact of OM is significantly moderated by the MRI tools. ESG and sub-ESG rating index are significantly influenced by the product's intrinsic characteristics. The coefficient of the household size variable shows that household size has a significant impact on the ESG or sub-ESG performance of OM.
add future directions:
As with any study, this research is not without limitations. Firstly, the current study is limited by the small number of cases analysed and the lack of case studies of ESG performance across different cultural contexts and industries. In addition, the ESG rating index is potentially affected by consumers’ psychology [63], which is not discussed in this paper due to the limitation of dataset structure. Overall, this study checks the two hypotheses proposed, explores and demonstrates the influencing mechanisms of ESG rating index in economic and business activities, and provides a fresh perspective based on traditional research. The formation of sub-ESG performance of accessible green foods (e.g. OM products) is further analysed, providing a strong basis for the construction of ESG evaluation system for the OM industry.
Based on the above analysis, the future direction of related research includes two main points. On the one hand, the analysis of different types of companies by expanding case sources will explore the generalisability and practical application of ESG evaluation systems. On the other hand, consumer psychology is introduced as an explanatory variable and ESG performance of OM producers will be further analysed in conjunction with market survey data to reveal the ESG-psychosocial impact of OM products.
Thank you very much for your suggestions and patient guidance, and we have revised and improved the relevant issues. Looking forward to hearing from your response.
Thank you and beat regards.
Sincerely,
Jiangyuan Hou
School of International Trade and Economics, University of International Business and Economics, No. 10 Huixin East Road, Chaoyang District, Beijing, 100029, China
Yanping Wang
School of Economics, Henan University of Economics and Law, No. 108 Jinshui East Road, Zhengzhou, He-nan, 450011, China
Mingyue Du
School of Economics, Beijing Technology and Business University, No. 11 Fucheng Road, Haidian District, Beijing, 100048, China

Reviewer 3 Report
Thank you for the opportunity to review your paper. I did not find any critical elements that need to be changed, and found it to be a well written and straightforward paper.
I had no issue with the language.
Author Response
Dear reviewer,
Thank you so much for your approve.
Best,
Reviewer 4 Report
sustainability-2475860
I am delighted to submit my review for the manuscript titled: The ESG and sub-ESG Impact of Consumers’ Socioeconomic Status and Product Attributes: Evidence from the China’s Market of Organic Milk. Please address these concerns and recommendations carefully.
1. In order to establish the significance and value of the study, it is necessary to provide a comprehensive rationale for the research, which emphasizes its relevance and unique contributions to the current scholarly discourse. This will strengthen the study's originality and scholarly impact.
2. Please revise the title and make it attractive for the readers.
3. Please use full abbreviation when using first time in text such as ESG.
4. The author must revise the abstract to include such gaps, how they fill them, topic importance, methods and clear-cut findings, and policy implications from research findings.
5. The introduction is generally acceptable for the first part, but it fails to establish the need for carrying out this study? What was the driving force behind conducting this particular study?
6. At the end of the introduction, in presenting the paper goals, try to answer the questions: How is the current research important? And how is it novel and contributes to the state of the art?
7. The research gap should be clearly defined.
8. What is your conclusion for theory and method development? Please, reflect on the conclusion. You need to convince the reader why this research is important and what its contribution.
Author Response
Responses to Reviewers’ Comments
Dear reviewer,
We are very grateful to the reviewers for their professional comments on our articles. These comments were very valuable and helped us to improve the quality of the manuscript by revising any problems in the article. Based on the suggestions of the comments, we have refined the framework structure of the article to present the research objectives, research questions, research innovations and research conclusions as clearly as possible. The linkage between the theoretical framework and the subsequent empirical evidence in the second subsection has been strengthened to make the argument more convincing. The data analysis is relatively well developed, but given the limitations of the data, we have also proposed the expected research direction of adding more cases and introducing consumer psychology. In the revised version of the document, changes to the manuscript are highlighted in green. Below are detailed responses to specific questions from various reviewers:
- 1. In order to establish the significance and value of the study, it is necessary to provide a comprehensive rationale for the research, which emphasizes its relevance and unique contributions to the current scholarly discourse. This will strengthen the study's originality and scholarly impact.
Answer:
Thanks for the reviewer’s advice. In the introduction, based on the theory of EPB and ECB, this research achieves these targets via the following path. Firstly, this study delves into the mechanisms by which consumers' socio-economic status and product attributes influence the ESG rating index, rather than simply analysing its impact on social responsibility performance. Secondly, the moderating variable of marketing resource intensity (MRI) is introduced into our econometric analysis to link the marketing tools into the ESG performance of OM producers. Thirdly, the hedonic analysis is conducted for the sub-ESG rating index, and the importance of environmental disclosure index is argued.
- Please revise the title and make it attractive for the readers.
Answer:
We have revised the title to read: Friend or Foe: How the consumers and producers affect ESG rating index? Evidence from the China’s market of organic milk.
- Please use full abbreviation when using first time in text such as ESG.
Answer:
In the second subsection, we point out the actual definition of ESG and show the full name of ESG issues as: environmental, social, and corporate governance issues.
- The author must revise the abstract to include such gaps, how they fill them, topic importance, methods and clear-cut findings, and policy implications from research findings.
Answer:
We have revised the abstract to clearly state the purpose of the study, the status of the study, the innovative nature of the study and the conclusions of the study. Specifically, this study attempts to investigate how product attributes and consumer socio-economic status affect OM product ESG performance. There aren't many pertinent studies on the interaction between the EGS evaluation system and organic milk products. Thus, this paper develops a targeted hedonic-regression model based on the EPB and ECB theories to investigate the ESG and sub-ESG impact of the above attributes based on the consumption dataset from the Kantar Worldpanel. We also introduce the moderating variable of marketing resource intensity (MRI) to check the influencing mechanism of sub-ESG rating index. This study demonstrates that the intensity of marketing resources has the moderating effect on the ESG performance of organic milk producers. Moreover, we also evidence that the household’ s family size and internal product attributes have a more significant impact on the ESG rating index. In addition, compared to social and governance information, the environmental information disclosure index has a more significant influence on the consumers’ socioeconomic status and product attributes compared with social and governance rating index. This study not only contributes to deepening the understanding of ESG evaluation system, but also provide a scientific guidance for the producers’ ESG strategy. Therefore, to increase the competitive advantage of OM enterprises, they should endeavor to construct the comprehensive evaluation system of entertainment social governance rating index and engage in the environmental investment activities.3.5 The introduction is generally acceptable for the first part, but it fails to establish the need for carrying out this study? What was the driving force behind conducting this particular study?
- The introduction is generally acceptable for the first part, but it fails to establish the need for carrying out this study. What was the driving force behind conducting this particular study?
Answer:
In the second subsection, we have revised the relevant content to identify the gaps in previous scholarly research in this area. Previous scholars have mainly studied the impact of consumers' socio-economic status and product attributes on corporate social responsibility performance, but the question of how relevant variables affect ESG rating indices is rarely discussed in the economic and business literature. Actually, the product quality and consumers’ socioeconomic features can potentially affect the ESG strategy of producers. This study lies in filling this gap to some extent, using organic milk products as an example. This study not only contributes to deepening the understanding of ESG evaluation system, but also provide a scientific guidance for the producers’ ESG strategy.
- At the end of the introduction, in presenting the paper goals, try to answer the questions: How is the current research important? And how is it novel and contributes to the state of the art?
Answer:
In the second subsection, we have revised the relevant content to identify the gaps in previous scholarly research in this area. Previous scholars have mainly studied the impact of consumers' socio-economic status and product attributes on corporate social responsibility performance, but the question of how relevant variables affect ESG rating indices is rarely discussed in the economic and business literature. Actually, the product quality and consumers’ socioeconomic features can potentially affect the ESG strategy of producers. This study lies in filling this gap to some extent, using organic milk products as an example. This study not only contributes to deepening the understanding of ESG evaluation system, but also provide a scientific guidance for the producers’ ESG strategy.
- The research gap should be clearly defined.
Answer:
In the theoretical framework section, we elaborate on the differences from previous scholars' research: Firstly, this study delves into the mechanisms by which consumers' socio-economic status and product attributes influence the ESG rating index, rather than simply analysing its impact on social responsibility performance. Secondly, the moderating variable of marketing resource intensity (MRI) is introduced into our econometric analysis to link the marketing tools into the ESG performance of OM producers. Thirdly, the hedonic analysis is conducted for the sub-ESG rating index, and the importance of environmental disclosure index is argued.
- What is your conclusion for theory and method development? Please, reflect on the conclusion. You need to convince the reader why this research is important and what its contribution.
Answer:
In the last subsection, we have modified the content to clearly state the findings of the study:
Theory and method development:
We demonstrate that consumer socio-economic characteristics and product attributes, moderated by the MRI, have a positive impact on the ESG rating index. Moreover, the environmental disclosure index plays a more positive role compared to the social and governance disclosure index. The findings of the study can intuitively guide the determination of corporate marketing strategies and the establishment of competitiveness improvement measures, and contribute to the construction of ESG comprehensive evaluation system and the implementation of ESG tools.
marketing implications:
Firstly, marketing to consumer groups with a high socioeconomic status can be chosen, for instance, in the organic milk market, where the econometric analysis shows that the online purchasing environment is superior to the offline environment. Secondly, the introduction of the MRI tool can better regulate the corporate marketing process. The findings also suggest that diversified marketing tools and differentiated product design can improve the company's competitiveness and achieve long-term profitability. Thirdly, the study of the sub-ESG disclosure index illustrates the important impact of environmental factors, and green product design and transparent environmental information disclosure can effectively contribute to the improvement of the company's brand image.
Thank you very much for your suggestions and patient guidance, and we have revised and improved the relevant issues. Looking forward to hearing from your response.
Thank you and beat regards.
Sincerely,
Jiangyuan Hou
School of International Trade and Economics, University of International Business and Economics, No. 10 Huixin East Road, Chaoyang District, Beijing, 100029, China
Yanping Wang
School of Economics, Henan University of Economics and Law, No. 108 Jinshui East Road, Zhengzhou, He-nan, 450011, China
Mingyue Du
School of Economics, Beijing Technology and Business University, No. 11 Fucheng Road, Haidian District, Beijing, 100048, China

Round 2
Reviewer 2 Report
No comment
Reviewer 4 Report
Authors have incorporated all my comments carefully. The paper can be accepted for publication.